# Leaving the health workforce during the COVID-19 pandemic: A cross-sectional study among Filipino healthcare workers

Christl Jan S. Tiu[1,2*], Nicole Rose I. Alberto[1,2], Maria Beatriz C. Baron[1], Michael Vincent V. Mercado[2], Ronnie E. Baticulon[3]

**1** College of Medicine, University of the Philippines Manila, Manila, Philippines, **2** Philippine General Hospital, Manila, Philippines, **3** Department of Anatomy, College of Medicine, University of the Philippines Manila, Manila, Philippines

\* cstiu1@up.edu.ph

## Abstract

The COVID-19 pandemic exacerbated the shortage of healthcare workers (HCWs), particularly in low- and middle-income countries. This cross-sectional descriptive study aimed to examine why hospital-based HCWs in the Philippines quit from their workplace at a time when HCWs were urgently needed to provide health services. Using an online, self-administered questionnaire distributed through personal and professional networks, we surveyed hospital-based Filipino HCWs who resigned between March 11, 2020, and September 15, 2021. We obtained demographics, workplace information, reasons for quitting, and proposed interventions. Among 70 valid responses, most of the HCWs were single (74%), female (59%), and without children (81%). More than half were nurses (31%) and physicians (29%). Most participants had one to five years of work experience (71%), worked in level 3 hospitals (70%), and had a schedule that required them to report for duty shifts for more than 8 hours (76%). While the majority of the HCWs were worried about the COVID-19 pandemic due to the risk of infection, the most frequent reasons for quitting were work overload/burnout, stress, and insufficient salary. To encourage workforce retention, the HCWs proposed increasing monetary compensation and non-salary incentives, cultivating a positive workplace culture, ensuring a reasonable workload, providing clear opportunities for career advancement, and improving workplace safety. These findings can guide ministries of health, policymakers, hospital administrators, health worker unions, and other stakeholders when planning for and responding to national or global health crises.

## Introduction

The World Health Organization (WHO) reported that the COVID-19 pandemic worsened the preexisting shortage of healthcare workers (HCWs) globally, especially in

**Data availability statement:** The de-identified dataset from this study is available for researchers upon request. To access the data, please submit a written request to the ethics board of the University of the Philippines Manila (e-mail: upmreb@post.upm.edu.ph; telephone: +63-2-8526436). Requests will be reviewed and approved in accordance with the research ethics committee's guidelines. Please note that the combined information of participants may be identifying, and therefore, only information that is not identifying in combination is available for external researchers. The results of this study have been used for publication and presentation, and further requests for raw data will be considered by the ethical board on a case-by-case basis.

**Funding:** The author(s) received no specific funding for this work.

**Competing interests:** The authors have declared that no competing interests exist.

low- and middle-income countries [1–3]. The Philippines is a lower-middle income country in Southeast Asia (Total population: 117 million in 2023 [4]) widely known for training HCWs, many of whom migrate to other countries [5–8]. This has resulted in chronic shortage of HCWs locally [8–10] despite many efforts to augment the health workforce, such as return service agreements where aspiring health professionals can avail of educational incentives and scholarships in exchange of working locally for at least 3 years [11–13]. In 2021, there were an estimated 7.9 doctors and 47.5 nurses and midwifery personnel for every 10,000 people in the Philippines [14]. Although the recent figures have surpassed the WHO-recommended threshold of 45 skilled health workers for every 10,000 people [15], most of the country's health workforce remains concentrated in cities and urban areas [9].

Many studies that had previously described the attrition of the health workforce during the COVID-19 pandemic focused on turnover intention [16,17]. Turnover intention, often used interchangeably with intention to leave, is defined as the individual's willingness to leave their current employment for a different position in the same or different profession [18–20]. During the pandemic, commonly cited reasons why HCWs considered leaving their workplace were related to moral distress. Examples of these include increased workload despite being short-staffed, insufficient personal protective equipment, allocation of scarce resources, fear of spreading COVID-19 to loved ones, and lack of organizational or government support [17]. Fear of COVID-19 was noted to be prevalent in both international and local studies [16,17,21,22]. To our knowledge, there is no published study that has documented the profiles and circumstances of HCWs who ultimately resigned.

In this paper, we aimed to describe the sociodemographic characteristics and work circumstances of Filipino HCWs who resigned from their hospital-based employment during the acute phase of the COVID-19 pandemic. Understanding the underlying reasons of HCWs who resigned can provide valuable insights for ministries of health, policymakers, public and private health administrators, third-party payers, and health worker unions, who must work together to encourage workforce retention and ensure continuity of health services during national and global health crises.

## Methods

### Ethics statement

The University of the Philippines Manila Research Ethics Board granted ethical approval (UPMREB Code 2021-523-01). Informed consent was obtained in the initial section of the online, self-administered, survey questionnaire.

This is a cross-sectional descriptive study using an online, self-administered survey in GoogleForms (Google LLC, Mountain View, California). We adhered to the Checklist for Reporting Results of Internet E-Surveys (S1 Checklist).

We defined HCWs as individuals who provide professional, health-related services to patients. These include physicians, nurses, occupational therapists, physical therapists, respiratory therapists, speech pathologists, pharmacists, medical technologists, and radiologic technologists, among others. For this study, we recruited

those who had worked for at least one year and resigned from their hospital-based employment between March 11, 2020 and September 15, 2021, which covered the Alpha and Delta waves of the COVID-19 pandemic in the Philippines. The study excluded HCWs who were on leave of absence or those who had been relieved from their positions. We collected responses from October 15 to December 30, 2021. We discontinued data collection due to a lack of new responses.

The survey questionnaire (S1 Text) collected data on the respondents' sociodemographic characteristics, workplace information and culture, worries of HCWs during the COVID-19 pandemic, reasons for leaving their positions, and proposed interventions for retention.

To assess the workplace culture, we used the Culture of Care Barometer [23]. Although originally designed to facilitate team exploration of workplace strengths and weaknesses, we used the Culture of Care Barometer in this study to investigate the culture of the respondents' previous workplace settings. Respondents rated items in this section using a Likert scale ranging from Strongly disagree to Strongly agree. Items were categorized into the following themes: Resources, Values, Team, Engagement, Empowerment, Role, and Management/Leadership.

To assess the degree and content of the worries of HCWs on COVID-19, including the perceived importance of psychological support, we adapted questions from a study by Sahashi et al. [24]. To explore reasons for quitting, we asked the respondents to choose reasons from a list, irrespective of priority, with an option for a free-form response. Finally, we asked the participants to describe, in a free-form response, any specific intervention that would convince them to return to work or remain as a hospital HCW during a pandemic.

We pilot-tested the survey among eight HCWs. We revised the survey questions based on feedback to improve the clarity of questions and to address potential issues in question construction. We distributed the revised survey questionnaire through social media platforms including Twitter, Facebook, and Instagram. We reached out to both personal networks and professional organizations to distribute the survey link among their connections (S1 List). Participation in the survey was open to all healthcare professionals with a registered email address. Using the first section of the online form, informed consent was obtained prior to answering the survey questionnaire. Upon completion, the participants could review their responses before submitting. Further, a copy of their responses was sent via their registered emails. All data collected were stored in the University of the Philippines Network and were only accessible to the authors.

After data collection, we exported de-identified survey responses to RStudio 2024.04.2 + 764 (Posit Software, PBC, Boston Massachusetts) for data analysis. We calculated the median and interquartile range for continuous variables and frequencies and percentages for categorical variables. Likert-scale responses were also reported as frequencies and percentages. Reasons for quitting were tabulated, creating categories as necessary for free-text responses. Finally, we conducted a thematic analysis of the proposed interventions to encourage workforce retention. Three authors independently analyzed and performed axial coding of the submitted free-text responses. Afterward, the authors convened to discuss the codes and to identify common themes, with the senior author resolving any disagreements.

## Results

### Survey respondents

We received a total of 78 responses, of which 70 (90%) were considered valid. Reasons for exclusion included the following: did not work in a hospital; occupation did not involve patient care; less than one year of experience as an HCW; and last day of work before March 11, 2020. Table 1 shows the sociodemographic characteristics of the survey respondents. The median age was 28 (IQR 7). Most of the participants were females (59%), single (74%), without children (81%), not the primary breadwinners of their families (71%), and had another household member contributing to the family income (80%). The most common jobs were nurses (31%) and physicians (29%). The majority had one to five years of experience in healthcare (71%). Nearly half of the respondents were from the National Capital Region (NCR, 46%).

 

**Table 1. Sociodemographic Characteristics of Survey Respondents.**

| Category | Subcategory | Frequency (n = 70) | Percentage (%) |
|---|---|---|---|
| **Age** | 20-24 | 7 | 10% |
| | 25-29 | 37 | 53% |
| | 30-34 | 14 | 20% |
| | 35-39 | 8 | 11% |
| | 40-44 | 2 | 3% |
| | 50-55 | 2 | 3% |
| **Gender** | Male | 24 | 34% |
| | Female | 41 | 59% |
| | Non-binary | 5 | 7% |
| **Civil Status** | Single | 52 | 74% |
| | Married | 18 | 26% |
| **With Children** | Yes | 13 | 19% |
| | No | 57 | 81% |
| **Primary Breadwinner** | Yes | 20 | 26% |
| | No | 50 | 74% |
| **With other household members contributing to household expenditures** | Yes | 56 | 80% |
| | No | 14 | 20% |
| **Location** | National Capital Region | 32 | 46% |
| | Cordillera Administrative Region | 1 | 1% |
| | Region I | 1 | 1% |
| | Region III | 3 | 4% |
| | Region IV-A | 8 | 11% |
| | Region V | 3 | 4% |
| | Region VI | 8 | 11% |
| | Region VII | 1 | 1% |
| | Region X | 3 | 4% |
| | Region XI | 7 | 10% |
| | Region XII | 1 | 1% |
| **Occupation** | Nurse | 22 | 31% |
| | Physician | 20 | 29% |
| | Medical technologist | 8 | 11% |
| | Physical therapist | 8 | 11% |
| | Pharmacist | 6 | 9% |
| | Radiologic technologist | 3 | 4% |
| | Respiratory therapist | 3 | 4% |
| **Years practicing as healthcare worker** | 1–5 | 50 | 71% |
| | 6–10 | 12 | 17% |
| | 11–15 | 6 | 9% |
| | 16–20 | 0 | 0% |
| | 21–25 | 1 | 1% |
| | 26–30 | 1 | 1% |

## Workplace information, culture, and worries

The number of respondents who used to work in the private sector were slightly higher than those in the public sector (54% vs. 46%, respectively). There was a lack of responses from pharmacists, physical therapists, radiologic technologists, and respiratory therapists in the public sector. The majority (70%) were employed at level 3 hospitals, defined as teaching hospitals with intensive care units, a rehabilitation unit, an ambulatory surgical clinic, a dialysis unit, a blood bank, a tertiary clinical laboratory, and an imaging facility with interventional radiology [25]. The top three areas where they worked were the emergency department (23%), ward (21%), and outpatient clinic (16%). Most of the participants (75%) worked on shifts that lasted more than eight hours per day. Half (50%) had > 10 patients served per shift (S1 Table). Physical therapists had the lowest median net salary at Php 14,000 (USD 1 = Php 49.62, 2020; IQR: Php 3,088) while doctors had the highest median net salary at Php 46,000 (IQR: Php 38,500) (S2 Table). Physicians and nurses from the public sector have median salaries that are higher than their counterparts from the private sector (Php 50,000 vs. Php 18,000 for physicians and Php 26,000 vs. Php 17,000 for nurses) (S3 Table). However, the small sample size does not allow for formal statistical testing.

The Culture of Care Barometer suggested that the study respondents had a neutral to positive workplace culture, as illustrated in Fig 1. More than eighty percent of participants agreed with the statements "When things get difficult, I can rely on my colleagues", "I feel respected by my co-workers", and "The people I work with are friendly".

On the other hand, almost half of the participants disagreed with the following statements: "The administration listens to staff views", "Hospital managers know how things really are", "There is strong leadership at the highest level in the hospital", "I feel supported to develop my potential" and "I get the training and development I need".

Fig 2 illustrates the Likert scale responses regarding worries about COVID-19. Most of the respondents agreed that they worried about the pandemic (90%), with the statement "risk of infection of family members and other relatives" being their primary concern (95%) (S4 Table). Less than half of the HCWs had either contracted COVID-19 themselves or had a family member that got infected. Nearly all (90%) reported having coworkers who had been infected.

## Reasons for quitting

The top three reasons for quitting were work overload and burnout (67%), stress (61%), and insufficient pay (59%) (Fig 3). Most participants made the decision to quit independently, without influence from their families (59%). Nearly half of the participants had already been contemplating leaving their jobs before the pandemic, but none planned to retire from the healthcare profession before 2025. Notably, many were planning to migrate and work abroad (49%). After resigning from their original employment, only a third remained working in healthcare settings locally (36%). The other respondents transitioned to nonclinical roles (29%), remained unemployed (17%), pursued employment abroad (9%), or pursued further studies (7%).

## Potential interventions to promote retention

The suggested interventions to promote HCW retention could be categorized under the following themes: increased monetary compensation, enhanced non-salary incentives, positive workplace culture, reasonable workload, clear opportunities for career advancement, and improved workplace safety. Representative responses from the HCWs are shown in Table 2.

## Discussion

To our knowledge, this is the first and only study to examine the characteristics and reasons of HCWs who quit from hospital-based employment during the COVID-19 pandemic in the Philippines. Our study found that fear of COVID-19 only ranked fifth among their cited reasons for quitting. The primary reasons for leaving the health workforce were preexisting issues, particularly work overload/burnout, stress, and insufficient pay.

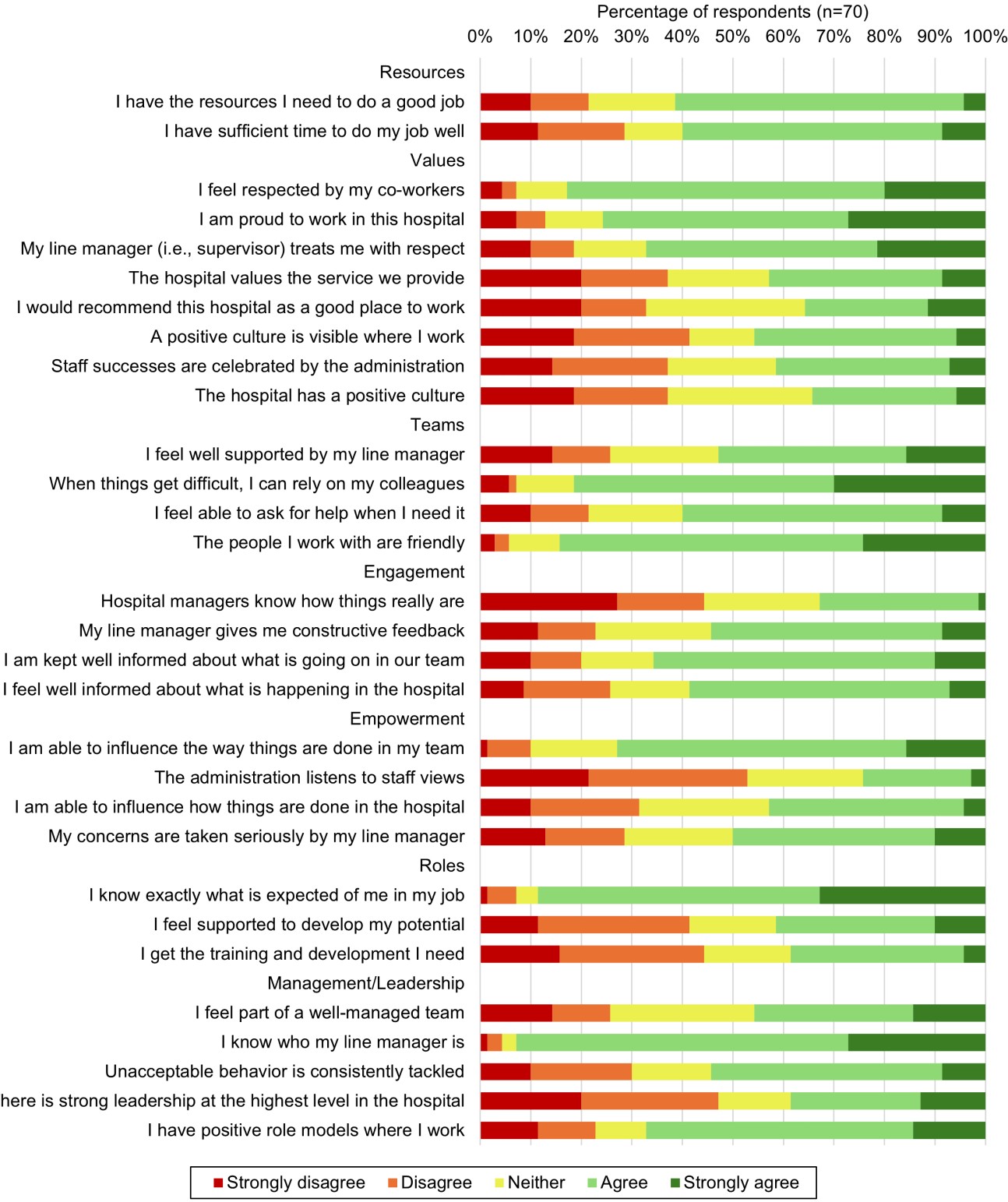

**Fig 1. Summary of Likert Responses on the Culture of Care Barometer.**

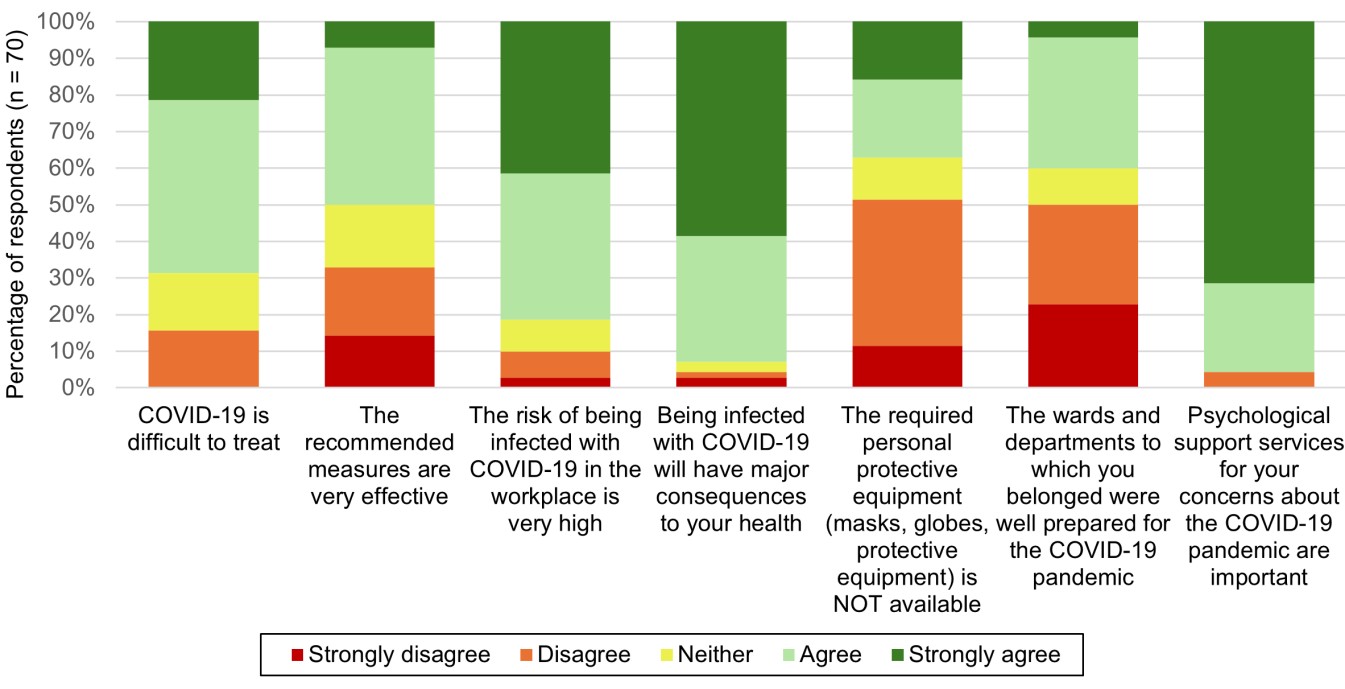

**Fig 2. Summary of Likert scale responses on worries about the COVID-19 pandemic.**

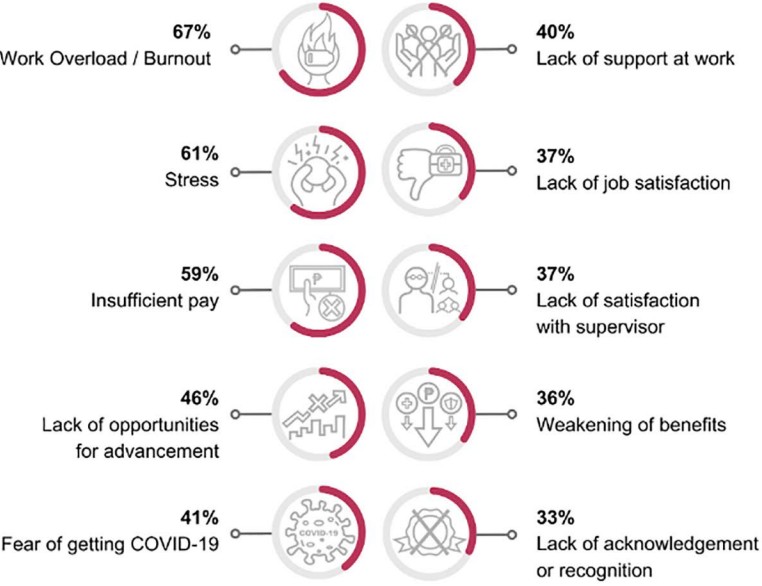

**TOP TEN REASONS FOR QUITTING**

67% Work Overload / Burnout

40% Lack of support at work

61% Stress

37% Lack of job satisfaction

59% Insufficient pay

37% Lack of satisfaction with supervisor

46% Lack of opportunities for advancement

36% Weakening of benefits

41% Fear of getting COVID-19

33% Lack of acknowledgement or recognition

**Fig 3. Top Ten Reasons for Quitting.** Created in BioRender. Tiu, C. (2025) https://BioRender.com/ivwgepr.

Global Public Health PLOS

**Table 2. Summary of interventions suggested to promote retention in local healthcare work.**

| Category | Intervention | Sample responses from healthcare workers |
|---|---|---|
| Increased Monetary Compensation | Increased Wages | "Increase basic salary" [HCW #39] |
| | Increased Hazard Pay | "Have better benefits like justifiable hazard pay" [HCW #20] |
| Enhanced non-salary Incentives | Provision of health insurance | "Support from supervisors and health insurance" [HCW #2] |
| | Provision of transportation | "Safe public transportation" [HCW #74] |
| | Provision of retirement plan | "Increased basic pay and benefits with retirement plan" [HCW #19] |
| | Provision of lodging | "Provision of bedspace for HCW" [HCW #23] |
| Positive workplace culture | Support from administration and coworkers | "Better support from administration, understanding and empathic coworkers" [HCW #12] |
| | Recognition and appreciation | "…recognition from the institution…" [HCW #70] |
| Reasonable workload | Balanced workload | "…adequate rest time and working hours." [HCW #40] |
| | Flexible schedule | "Flexible work hours" [HCW #34] |
| | Adherence to work shift limitations | "No mandatory overtimes" [HCW #56] |
| | Shorter shifts | "Additional workforce to shorten shift/number of hours HCW must report to work" [HCW #13] |
| | Additional healthcare workforce | "… improve nurse-patient ratio for quality nursing care" [HCW #66] |
| Clear opportunities for career advancement | Opportunity for continuing education | "… opportunities for career growth – more nursing specialization" [HCW #24] |
| | Equal opportunities for promotion | "… opportunity to grow, *walang palakasan system* in promotion *[no nepotism in promotion]*" [HCW #30] |
| | Mentoring from superiors | "…guidance from mentors…" [HCW #4] |
| Improved workplace safety | Provision of personal protective equipment | "…ample provision of PPEs" [HCW #55] |
| | Provision of free COVID-19 test | "…free swabbing…" [HCW #57] |
| | Increased vaccination coverage | "…improvement in vaccination rates." [HCW #43] |

Fear of COVID-19 is the most prevalent theme for turnover intention among HCWs [17]. Yet, our findings suggest that while possibly contributory, being infected with COVID-19 is unlikely to be the primary reason for quitting. Notably, most of our respondents and their family members have not contracted the virus. Additionally, over 70% of HCWs in a study at the Philippine General Hospital – many of whom have previously contracted the virus – stated willingness to work throughout the COVID-19 pandemic [26].

Affirming the findings of Poon, et al., our study found that the worsening of adverse working conditions contributed to the attrition of HCWs [17]. Many of our participants quit work due to the overwhelming workload brought on by the pandemic. They mostly worked from level 3 hospitals and in patient-intensive areas such as the emergency department, wards, and outpatient services. Further, majority also reported working beyond the prescribed 8-hour working shift by the Labor Code of the Philippines [27]. The increase in workload and long work hours likely intensified burnout and stress among those who quit. A qualitative study of nursing home staff in the United States reported similar findings, where,

according to the respondents, the COVID-19 pandemic significantly increased staff workload, work hazards, and financial strains while decreasing job satisfaction and influencing staff turnover [28].

Despite working long hours and heavy workloads in high-risk environments, the participants reported a low net salary. During the pandemic, the government introduced additional compensation benefits such as COVID-19 hazard pay, special risk allowance, and comprehensive health insurance coverage for COVID-19-positive HCWs admitted to hospitals [29–32]. However, HCWs from the private sector were not uniformly covered and there were delays in providing these benefits [33,34]. Further, these benefits did not apply to family members who were also at risk of getting infected. Thus, being infected or having a family member infected and hospitalized could lead to impoverishing or catastrophic health expenditure.

To illustrate, in the Philippines, the estimated average out-of-pocket costs for being admitted for COVID-19 ranged from Php 4,005.60 to 44,428.63 [35]. The nurses who participated in the current study earned a median salary of Php 22,000 monthly. This meant that if a nurse or a family member were to get sick and admitted for COVID-19, the cost would potentially be equivalent to 2 months of work. That being said, the range provided for out-of-pocket hospitalization cost was likely an underestimation because this range was sourced from a public, tertiary-level hospital in the Philippines. This did not reflect hospitalization costs at private institutions and did not account for indirect costs such as lost wages.

The findings in this study contribute to literature explaining why there is an exacerbation of HCW shortage despite having a surplus of professionals in the Philippines [8,10]. The first reason is the persistence of brain drain, in which HCWs train locally but seek better opportunities in other countries [36]. Most of our participants were young professionals at the early stages of their healthcare careers. Among those who disclosed their intention to leave before the pandemic, nearly half desired to migrate and work abroad. The second reason is the transition of HCWs to other professions. Consistent with previously published findings by the Professional Regulatory Commission of the Philippines, only a portion of registered health professionals would eventually practice as HCWs [37]. In the current study, many participants transitioned to non-clinical roles, continued their education, or remained unemployed.

Recommendations to promote health workforce retention have remained consistent with pre-pandemic demands. This highlights longstanding issues in human resources for health. Even before the pandemic, a systematic review published in 2008 by Willis-Shattuck et al. [38] documented that significant motivators for HCW retention in low- and middle-income countries included financial rewards, career development, continuing education, hospital infrastructure, resource availability, hospital management, and recognition/appreciation – most of which were also proposed by our study participants. Additional suggestions such as the availability of personal protective equipment to ensure work safety have also been documented to increase willingness to work during health emergencies [39].

In light of our findings, it is important to take proactive steps to address the longstanding issues in the health workforce. The World Health Organization has provided a global strategy to address challenges in the health workforce. This includes creation and implementation of policies to address migration, bringing back workers into the health sector, and attracting unemployed health workers [2]. Rather than being reactive to HCW turnover, stakeholders must work together and invest in sustainable solutions to address existing problems such as work overload, burnout, and insufficient pay to encourage the retention of HCWs locally, not only in response to health emergencies, but more important, as a long-term public health investment.

## Limitations

The online dissemination of the survey via convenience sampling on social media platforms and personal/professional networks limited the participant pool to those with social media accounts, internet access, and familiarity with using digital devices. Participant recruitment was conducted on a voluntary basis, introducing the potential for selection bias. To mitigate this, the survey was promoted through the social media accounts of professional organizations to achieve a broader reach. Geographic locations with limited internet access were not adequately represented and should be further explored

in future research. The present study includes a disproportionately large number of participants from the NCR. As a result, our findings may have limited generalizability to other regions, which may have distinct experiences and challenges. Future research could seek a more proportionate representation across regions to capture the perspectives of HCWs beyond the NCR and enhance the generalizability of findings. Furthermore, the present study does not have a proportionate sample across HCW occupations from the private and public sectors. Considering that there may be differences between the two sectors (e.g., salary), obtaining a proportionate sample of HCWs from each occupation within private and public hospitals may offer a more generalizable and comprehensive understanding of the factors influencing each sector.

Ideally, a validated tool would have been used to investigate this issue. However, due to the acuity and time-dependent nature of data collection, we chose to adapt questionnaires from other studies covering the concepts of interest. Larger sample size and in-depth interviews to capture qualitative data would have contributed further to the current research. However, recruitment may be challenging as HCWs who resigned do not have organized groups. Future research may explore whether the HCWs who quit during the COVID-19 pandemic returned to work or retired from healthcare completely.

## Conclusion

In this study among Filipino HCWs who resigned from their hospital-based employment during the early phase of the COVID-19 pandemic, we found that preexisting issues remained the most important reasons for quitting: work overload, burnout, stress, and insufficient pay. To strengthen the health workforce and encourage workforce retention, it is essential that ministries of health, hospital administrators, professional organizations, and health worker unions work together, to ensure continuity of health services even during national and global health crises.

## Supporting information

**S1 Checklist. Checklist for Reporting Results of Internet E-Surveys (CHERRIES).**
(DOCX)

**S1 Text. Survey Questionnaire.**
(DOCX)

**S1 List. List of Organizations.**
(DOCX)

**S1 Table. Workplace information of survey respondents.**
(XLSX)

**S2 Table. Salary by Occupation.**
(XLSX)

**S3 Table. Salary by occupation, public vs. private.**
(XLSX)

**S4 Table. Causes of concern among healthcare workers.**
(XLSX)

## Acknowledgments

The authors would like to acknowledge Dr. John Carlo B. Reyes and Dr. Ian Kim Tabios who stood as advisors and provided inputs on the possible directions of the research in the initial phases.

## Author contributions

**Conceptualization:** Christl Jan S. Tiu, Maria Beatriz C. Baron, Ronnie E. Baticulon.

**Data curation:** Christl Jan S. Tiu, Nicole Rose I. Alberto, Maria Beatriz C. Baron, Michael Vincent V. Mercado.

**Formal analysis:** Christl Jan S. Tiu, Nicole Rose I. Alberto.

**Methodology:** Christl Jan S. Tiu, Nicole Rose I. Alberto, Maria Beatriz C. Baron, Michael Vincent V. Mercado, Ronnie E. Baticulon.

**Project administration:** Christl Jan S. Tiu, Michael Vincent V. Mercado.

**Resources:** Christl Jan S. Tiu.

**Supervision:** Ronnie E. Baticulon.

**Validation:** Maria Beatriz C. Baron.

**Visualization:** Christl Jan S. Tiu.

**Writing – original draft:** Christl Jan S. Tiu, Nicole Rose I. Alberto, Maria Beatriz C. Baron.

**Writing – review & editing:** Christl Jan S. Tiu, Nicole Rose I. Alberto, Maria Beatriz C. Baron, Michael Vincent V. Mercado, Ronnie E. Baticulon.

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
