## [Decision Letter · Decision Letter 0]

30 May 2025

PGPH-D-24-02821

Leaving the health workforce during the COVID-19 pandemic: A cross-sectional study among Filipino healthcare workers

Dear Dr. Tiu,

Thank you for submitting your manuscript to PLOS Global Public Health. After careful consideration, we feel that it has merit but does not fully meet PLOS Global Public Health’s publication criteria as it currently stands. Therefore, we invite you to submit a revised version of the manuscript that addresses the points raised during the review process, with acknowledgement of the limitations as detailed by reviewer 1 and the minor revisions as detailed by reviewer 2 below. 

We look forward to receiving your revised manuscript.

Kind regards,

Padmasayee Papineni, MD

Academic Editor

Journal Requirements:

Additional Editor Comments (if provided):

Reviewers' comments:

Reviewer's Responses to Questions

**Comments to the Author**

1. Does this manuscript meet PLOS Global Public Health’s publication criteria?

Reviewer #1: Yes

Reviewer #2: Yes

2. Has the statistical analysis been performed appropriately and rigorously?

Reviewer #1: Yes

Reviewer #2: Yes

3. Have the authors made all data underlying the findings in their manuscript fully available (please refer to the Data Availability Statement at the start of the manuscript PDF file)?

Reviewer #1: Yes

Reviewer #2: Yes

4. Is the manuscript presented in an intelligible fashion and written in standard English?

Reviewer #1: Yes

Reviewer #2: Yes

Reviewer #1: This study addresses an important and timely issue by examining the factors affecting attrition among Filipino healthcare workers. Filipino healthcare professionals play a critical role not only in the local healthcare system but also in filling healthcare workforce gaps globally. Understanding the factors driving their decision to leave the profession or migrate is essential for informing policies aimed at retention and workforce sustainability. The focus on this key demographic makes this study a valuable contribution to the field.

One notable limitation of the study is the disproportionate representation of respondents from the National Capital Region (NCR), which accounts for a significant portion of the sample. Given that the NCR is not proportionate to the overall population of healthcare workers in the Philippines, this geographic imbalance may limit the generalisability of the findings to healthcare workers in other regions, who may face different challenges and circumstances. For instance, healthcare workers in rural areas might have distinct experiences related to resources, infrastructure, and access to support systems compared to their counterparts in urban hospitals like the NCR. To strengthen the validity of the study, it would be beneficial to address this sampling bias by including a more regionally representative cohort of participants or by discussing how this limitation might have influenced the results.

It would be helpful if the author examined how the cohort is distributed between healthcare workers in the public and private sectors. In the Philippines, there is a substantial salary gap between these sectors, which could influence attrition rates. Exploring whether attrition is higher among healthcare workers in public hospitals compared to private hospitals could provide valuable insights and may further contextualise the findings. Including this analysis or acknowledging its absence as a limitation would strengthen the study.

Reviewer #2: Reviewer comments to the author

The manuscript by Christl et al reports the results from a cross-sectional descriptive study that focused on why hospital-based HCW’s in the Philippines quit from their workplace during COVID-19 pandemic period.

The study is interesting; however, the below issue should be addressed by the authors.

1. Table 1 – Sociodemographic characteristics of survey respondents. Please review column 1-category on ‘Years practicing as healthcare worker’-subcategory under column 2, range 16 to 20 is missing.

**Do you want your identity to be public for this peer review?** For information about this choice, including consent withdrawal, please see our Privacy Policy

Reviewer #1: No

Reviewer #2: No

---

## [Editor Report · Decision Letter 1]

1 Oct 2025

Leaving the health workforce during the COVID-19 pandemic: A cross-sectional study among Filipino healthcare workers

PGPH-D-24-02821R1

Dear Dr Tiu,

We are pleased to inform you that your manuscript 'Leaving the health workforce during the COVID-19 pandemic: A cross-sectional study among Filipino healthcare workers' has been provisionally accepted for publication in PLOS Global Public Health.

Best regards,

Padmasayee Papineni, MD

Academic Editor